# Association Between the Activity Space Exposure to Parks in Childhood and Adolescence and Cognitive Aging in Later Life

**DOI:** 10.3390/ijerph16040632

**Published:** 2019-02-21

**Authors:** Mark P.C. Cherrie, Niamh K. Shortt, Catharine Ward Thompson, Ian J. Deary, Jamie R. Pearce

**Affiliations:** 1Centre for Research on Environment, Society and Health (CRESH), University of Edinburgh, Edinburgh EH8 9XP, UK; Niamh.Shortt@ed.ac.uk (N.K.S.); jamie.pearce@ed.ac.uk (J.R.P.); 2OPENspace Research Centre, University of Edinburgh, Edinburgh EH3 9DF, UK; c.ward-thompson@ed.ac.uk; 3Centre for Cognitive Ageing and Cognitive Epidemiology, Department of Psychology, University of Edinburgh, Edinburgh EH3 9DF, UK; iand@exseed.ed.ac.uk

**Keywords:** green space, road traffic accidents, cognitive aging, activity space, life-course perspectives, environmental exposures

## Abstract

The exposure to green space in early life may support better cognitive aging in later life. However, this exposure is usually measured using the residential location alone. This disregards the exposure to green spaces in places frequented during daily activities (i.e., the ‘activity space’). Overlooking the multiple locations visited by an individual over the course of a day is likely to result in poor estimation of the environmental exposure and therefore exacerbates the contextual uncertainty. A child’s activity space is influenced by factors including age, sex, and the parental perception of the neighborhood. This paper develops indices of park availability based on individuals’ activity spaces (home, school, and the optimal route to school). These measures are used to examine whether park availability in childhood is related to cognitive change much later in life. Multi-level linear models, including random effects for schools, were used to test the association between park availability during childhood and adolescence and cognitive aging (age 70 to 76) in the Lothian Birth Cohort 1936 participants (N = 281). To test for the effect modification, these models were stratified by sex and road traffic accident (RTA) density. Park availability during adolescence was associated with better cognitive aging at a concurrently low RTA density (β = 0.98, 95% CI: 0.36 to 1.60), but not when the RTA density was higher (β = 0.22, 95% CI: −0.07 to 0.51). Green space exposure during early life may be important for optimal cognitive aging; this should be evidenced using activity space-based measures within a life-course perspective.

## 1. Introduction

The neighborhood environment during early life may be important for mental health [1] and cognitive aging [2], given that this is a period of heightened brain plasticity in childhood [3] and behavioral development in adolescence [4] (see Figure 1 in [5]). Green space may play a role in cognitive development, with several studies finding an association between higher exposure to green spaces and better childhood academic performance [6,7,8] and social, emotional, and behavioral outcomes [9]. Evidence for the association between green space and improved cognitive development being mediated through lower air pollution has been presented [8,10], although other distinctive benefits to cognitive health have been proposed. Green space may have a positive effect on a child’s cognitive development by providing a space for physical activity [11] and by moderating the impact of life stress and adversity [12], as well as indirectly through reduced stress in the child’s caregiver [13]. Both increased physical activity and reduced stress could increase educational involvement and attainment throughout life and could, therefore, boost cognitive reserve; this, in theory, could create a buffer against a future decline in cognitive function in later life [14].

A number of studies have also found a greater availability of green space in adulthood to be associated with higher cognitive function [15]. Only one study has been longitudinal, spanning 14 years in later life and using a baseline measure of the green space contact (daily vs. rarely gardening) [16]. Given the difficulty of continuously measuring contact with the natural environment throughout life, there is a lack of evidence on whether the green space can have lifelong effects on cognitive aging. We previously created a measure of the lifelong exposure to green space (park availability) from childhood to later adulthood in areas across the Edinburgh region of Scotland. Using this measure, we found that the higher childhood residential-based park availability was associated with a slower rate of decline in cognitive function from age 70 to 76, conditional on the sustained availability in adulthood [2]. Whilst this life-course research was an important advance in terms of understanding how environmental circumstances in childhood can have lifelong implications for health, the approach relied on simplistic residential-based measures of green space exposure, based solely on the participants’ residential location. Exposure based on residential location can be prone to misclassification bias, due to the movement of people outside of their neighborhood for work, school, or recreation [17,18]. Areas where people travel to and from during their daily activities are termed the ‘activity space’. One previous study investigated the association between activity space greenness and the cognitive development in children and found that a higher level of greenness surrounding the school, home, and on the child’s route to school was associated with a greater improvement in the 12-month working memory and attentiveness [8]. 

This paper seeks to combine the approaches of these two studies [2,8] by using an activity space measure of the green space in the same life-course model of cognitive aging published previously [2]. In doing so, the paper simultaneously addresses key aspects of contextual uncertainty that have threatened to undermine the robustness of studies examining the connections between health and places. Firstly, we developed a measure of exposure that moved beyond simple residential estimates by also using information on the availability of green space (identified as the area of public parks within a buffer zone) nearby the school and the route to school. This more sophisticated measure of the exposure was then used in a life-course model as a predictor of the cognitive aging in later life [2]. 

The conceptual model shows that the childhood activity space can be defined by geographic locations around the home, primary and secondary school, recreation spaces, and the routes between these areas. The movement around these locations is influenced by demographic characteristics (e.g., age) and the parental perceptions of space (e.g., safety). Activity spaces for childhood life stages (e.g., infancy) determine the exposure to urban green space (e.g., public parks), which can influence healthy cognitive aging, after accounting for green space access in later life and socioeconomic status. 

While we may assume a certain spatial extent of mobility around certain points of interest (i.e., home and school), the actual movement is influenced by factors such as age, sex, and the parental perception of traffic safety [19]. We, therefore, hypothesized that being female [20], of a lower age, and living in an area with a higher number of Road Traffic Accidents (RTAs) (as a proxy for the parental perception of traffic safety [19]), would limit the activity space, reduce the exposure to green space, and attenuate associations with cognitive aging in later life. To test these hypotheses, we divided the estimates of the exposure to public parks into childhood (age 4–11) and adolescence (age 11–18) categories and used these as the primary predictors in a multi-level regression model. Our first aim was to determine whether the availability of public parks in childhood and adolescence was associated with cognitive aging later in life. Our second aim was to determine whether these associations were modified by the sex of the participant or by RTA levels in early life activity spaces. The conceptual model is detailed in Figure 1.

## 2. Materials and Methods

### 2.1. Study Design and Setting

The Lothian Birth 1936 (LBC1936), based within and around the British city of Edinburgh, follows a subgroup of community-dwelling older people, most of whom took a test of their general intelligence in 1947 (at age 11) as part of the Scottish Mental Survey 1947 (SMS1947; N = 70,805). The cohort participants were re-contacted between 2004 and 2007 if they were identified from the Community Health Index (GP-registered individuals) and/or through media adverts; from 2318 responses, 1091 were eligible for wave 1 data collection, at a mean age of 70 years [21,22,23]. The interviews were held at the Wellcome Trust Clinical Research Facility at the Western General Hospital in Edinburgh. To assess residential movement throughout life, the participants were asked to complete a decade-based “life grid” questionnaire (see Appendix A) at a mean age of 78 (July 2014–April 2015). The life grid questionnaire technique minimizes recall bias by encouraging the participant to input global and local events as memory prompts [24]. Participants in the LBC1936 at age 78 (N = 704) were requested to provide a residential address for each decade from the 1930s onwards. The completion of the life grid questionnaire was 84% (N = 593/704).

### 2.2. Measures of the Green Space and Road Traffic Accidents

The “Civic Survey and Plan for the City & Royal Burgh of Edinburgh” was published in 1949 and includes surveys of population, education, recreation, housing, green space, and traffic from the 1940s, as well as a number of associated maps of these data [25]. We digitized the area of public parks using Map 9 as our measure of childhood green space for the LBC1936 [24]. We also digitized a public park measure from adulthood (1969; the participant mean age of the LBC1936 participants was 33) using data collected by the Town Planning Department, Edinburgh [26]. Further details on these data and the strengths and limitations are presented elsewhere [24]. The road traffic accident point locations were available from 1937 and 1946 in Map 13 [25] and were georeferenced using ArcGIS. Kernel Density Estimation (KDE) was used to create a continuous surface based on the locations of RTAs. The KDE divides the area containing RTAs into 100 by 100 m cells, then provides a proximity-weighted (a 1500 m search radius was used as the hypothesized upper-end distance that a child might walk, i.e., a 30-min walking distance) estimate of the density of accidents per km^2^.

### 2.3. Linking the Measures of Parks and the Road Traffic Accident Density to the LBC1936

The participants were eligible for the current analysis if they had an Edinburgh-based address from 1936–1952 (N = 311) and at least one address during adulthood (N = 281). The locations they resided in for the longest duration between 1936 and 1952 were used as their home location so that each participant had only one childhood home address. The locations of primary and secondary schools in Edinburgh were identified from the survey of Abercrombie and Plumstead (1949) and were georeferenced, which we linked to the name of the school that the participant attended. We simulated an optimal route to school based on contemporary Google Maps (2017) walking directions using the “ggmap” package in R 3.3.2. To estimate the public park availability, we created a buffer (1000 m) surrounding the participant’s home, primary school, and secondary school. The intersection between the buffer and the public parks determined the availability (i.e., the area of the parks within the buffer as a percentage of the total buffer area). We created a buffer surrounding the route to school and the availability of public parks was assessed by the intersection, as above. Where the route and the home or school buffer overlapped, the overlap was discounted, to avoid overestimation. The road traffic accident density was estimated by using the values from the density surface at the home and school points. 

### 2.4. Childhood and Adolescence Activity Space-Derived Green Space and Road Traffic Accident Density

In summary, we gained eight measures of the environment for each participant, including the public park availability for the home, primary school, and secondary school, the route from the home to the primary school, the route from the home to the secondary school, and the RTA density around the home, primary school, and secondary school. 

From these measures, we built two public park indices: The childhood index (age 4–11), which consisted of the park availability at the home, primary school, and on the route to primary school; and the adolescence index (age 11–18), which consisted of the park availability at the home, secondary school, and on the route to secondary school. We weighted each variable in the index by the hypothesized average daylight hours (12 h) spent in each home (3 h), school (8 h), and on the route to school (1 h), as defined in a previous study [8]. In a sensitivity analysis, we created the childhood and adolescence indices for the following activity space buffers: Large (home: 1500 m, school: 1500 m, route: 300 m) and small (home: 500 m, school: 500 m, route: 100 m). The largest buffer size (1500 m) was determined by data on the typical distances traveled to city parks in the survey conducted in 1969, which found that approximately 90% of the 500 people surveyed traveled up to 1.6 km to reach public parks within the city [26].

Road traffic accident indices were constructed by applying the same temporal weighting as above so that we had RTA indices for childhood and adolescence to match with the public park indices. These indices were categorized as high and low by the median. All georeferencing was undertaken using ArcMap 10.1 GIS software (ESRI, Redlands, CA, USA) and the geoprocessing was undertaken in R 3.3.2 (R Foundation for Statistical Computing, Vienna, Austria). 

### 2.5. Outcome: Change in Cognitive Function

Age-standardized test scores for the Moray House Test no. 12 (MHT) were used as the measure of cognitive function at the mean ages of 70 and 76 [27]. This is the same test that participants had taken in the Scottish Mental Survey 1947 at the mean age of 11 years old. The MHT contains 71 items on a range of mental tasks (e.g., following directions, reasoning, arithmetic) and had a maximum score of 76. It is a paper-and-pencil test and there is a time limit of 45 min. The concurrent validity in childhood is shown by a correlation of about 0.8, with an individually-administered Terman-Merrill revision of the Stanford Binet test [27]. The MHT has concurrent validity in older age, as seen by a high correlation with the Wechsler tests of intelligence [28]. The change from age 70 to age 76 in cognitive function was determined by calculating the standardized residuals, which were predicted using a linear regression model with the latter time point as the dependent variable and the earlier time point as the independent variable. This change score was interpreted as the deviation from what was expected, given the prior test scores, so that a positive coefficient was a better than expected change and a negative coefficient was a worse than expected change. 

### 2.6. Covariates

The relationship between park availability and cognitive function could be affected by socioeconomic status [29]. Therefore, in the models, we used the father’s Occupational Social Class (OSC) as a covariate to represent the childhood socioeconomic status. This variable was created by classifying the father’s occupation into one of the following categories: I: “Professional”, II: “Managerial and technical occupations”, IIIN: Skilled occupations, non-manual”, IIIM: “Skilled occupations, manual”, IV: “Partly skilled occupations”, and V: “Unskilled occupations”. This was then dichotomized into professional-managerial (I, II) and skilled, partly skilled, unskilled (IIIN, IIIM, IV, V) categories. We also used the questionnaire response on the number of people per room in the childhood home, asked at age 70, to further account for the childhood socioeconomic status. For the socioeconomic status during adulthood, we used the participant’s OSC coded in the same way. We also used several variables relating to behavior (i.e., childhood smoking, adulthood smoking, and adulthood alcohol consumption), which are often associated with socioeconomically patterned geographical variables [30], proposed to be on the pathway between environmental variables and cognitive health [5]. Further details on the operationalization of covariates are provided elsewhere [2]. The percentage of missing information for covariates and outcomes was low (≤6%) and was assumed to be missing at random. We used multiple imputation by chained equations, using the “mice” R package with selected auxiliary variables known to be associated with the outcome and covariates, to create 20 complete datasets [2]. The results presented are the pooled estimates from these datasets. 

### 2.7. Statistical Analysis

We developed multi-level linear regression models, clustering participants within their schools, to account for the structure of the data. We used the participant’s primary or secondary school as a random effect. It was previously found that the relationship between the childhood park availability and later adulthood cognitive function is modified by the adulthood park availability [2], with this model being the best fitting out of a number of other candidate life-course models (Appendix A). Therefore, we used the participants’ home addresses from 1953–1989 (i.e., the mean of the park availability surrounding each home location) to create a multiplicative interaction variable (with the buffer size corresponding to the early life variable). These interaction variables were used as fixed effect predictors. A linear model was constructed, with the change in cognitive function from age 70 to age 76 as the dependent variable. We present the results adjusted for sex, childhood covariates (i.e., the father’s OSC, people per room in the childhood home, and childhood smoking), and adulthood covariates (adulthood OSC, alcohol consumption, and smoking status). We ran the models for the full cohort and then stratified by sex and road traffic accident density. In a sensitivity analysis, we used the Bonferroni adjustment for multiple comparisons. The statistical significance was set at *p* < 0.05. All statistical analyses were conducted using the ‘lme4’ package in R 3.3.2. 

## 3. Results

The distribution of the LBC1936 analysis sample for selected characteristics is presented in Table 1. There were slightly more men, more participants who had fathers in a “skilled, partly skilled or unskilled” OSC, and approximately equal proportions of the participants in a “professional-managerial” or “skilled, partly skilled or unskilled” OSC. The most common qualification was attained from high school (“O-level or equivalent” (47%)), followed by “No qualification” (15%). In comparison to the full sample of LBC1936 participants who supplied their residential address history (N = 592), the analysis sample (N = 281) had a higher number of participants in lower socioeconomic status and lower educational attainment categories (Appendix A). 

A slightly larger availability of parks was found during adolescence (9.1% ± 6.9) compared to childhood (8.6 ± 7.3) (Table 1). The road traffic accident density was much higher during adolescence (14.5 km^2^ ± 5.2) compared with childhood (6.9 km^2^ ± 3.6), due to secondary schools being close to the main thoroughfares through the city and primary schools being located in local, more residential, areas spread around the city (Figure 2). The road traffic accident density was concentrated in the center and North East (in the port of Leith), whereas public parks were more equally spaced, but with the greatest area close to the center due to the largest park in Edinburgh being located there. 

The availability of parks during adolescence was associated with better cognitive aging (β = 0.27, 95% CI: 0.00 to 0.55) to a greater extent, compared with the childhood availability of parks (β = 0.22, 95% CI: −0.07 to 0.51) (Table 2). The marginal effects of the park availability during childhood and adolescence, conditional on the availability of adulthood parks, is presented in Figure 3. This shows that the coefficient for the park availability during adolescence becomes increasingly positive (better cognitive aging) with an increasing adulthood park availability. The associations were slightly higher in the females compared to the males; males had a coefficient of 0.21 (95% CI: −0.20 to 0.62) and females had a coefficient of 0.33 (95% CI: −0.07 to 0.72), however, the female coefficient was not notably higher (Table 2). The park availability during adolescence had strong associations with better cognitive aging in participants with a lower RTA density (β = 0.98, 95% CI: 0.36 to 1.60); this was not the case when the RTA density was higher (β = 0.08, 95% CI: −0.29 to 0.45) (Table 2). In the sensitivity analysis with smaller and larger sized buffers, the childhood and the adolescent park availability were not associated with cognitive aging (Appendix A). In our case, applying a multiple testing correction would have meant that our main finding, i.e., adolescent park availability with a low RTA density, occurred by chance. 

This shows the marginal effects of the childhood park availability on cognitive change from age 70 to 76, conditional on the percentage of parks in adulthood. This shows that the childhood/adolescent park availability had an increasingly positive/advantageous association with the cognitive change from age 70 to 76 years, as the person’s adulthood park availability increased. 

## 4. Discussion

### 4.1. Principal Findings

The current study has used a novel approach of utilizing information on the home, school, and optimal route to school to estimate the park availability in early life and investigate the associations with cognitive aging in later life. Our current work builds on previous literature on everyday time-space interactions and their relation to experiences over the life-course [31]. A key novelty of this study is the integration of early life activity space-based measures within a life course framework, which addresses two important concerns identified in the literature on contextual uncertainty [18]. Public park availability for the adolescent activity space (home, school, route to secondary school) was positively associated with better cognitive aging in later life, especially for those living in low RTA density areas. This is in addition to residence-only models found in this sample previously [2]. Crucially, these associations were shown to be more robust to the adjustment of socioeconomic and behavioral variables than in the model using residence-only exposure, which did not hold after adjustment (β = 0.26, 95% CI: −0.06 to 0.57) [2]. This shows that the variation in the effect size of the association between green space and cognitive aging is partly dependent on how the geographical exposure is specified and operationalized, which is an example of the uncertain geographic context problem (UGCP) [17]. Recent work has emphasized the salience of the UGCP in the study of the built environment in general and green space in particular, on obesity [32,33]. Further work is required to investigate statistical and theory-driven strategies to determine an optimal measurement of activity spaces, which are specific to certain exposures and outcome relationships. 

### 4.2. Relation to Other Studies

The greater independence of mobility during adolescence may explain why the adolescent park availability seemed to be more important for cognitive aging than in childhood (i.e., the exposure estimates were more accurate). A potential mechanism behind this finding is that parks affect behavioral development in adolescence [34]. A recent study found that aggressiveness in adolescents was reduced with greater greenness within 1000 m from the home; the difference between the highest and lowest greenness exposures within the study equated to roughly 2–2.5 years of behavioral maturation [34]. We found that the results were non-significant with smaller (500 m) and larger (1500 m) sized buffers. In residence-only models previously, both 1000 m and 1500 m buffers were significant [2]. This indicates that activity space-based measures of green space are more sensitive to buffer sizes. We found support for a modification of the effect between park availability and cognitive aging by the RTA density. This could be explained by the activity space, and therefore the exposure to parks, being reduced for participants with a higher road traffic accident density. Both perceptions of road safety [35] and road safety features (i.e., more traffic lights) [36] were associated with a greater likelihood of walking and cycling, although this was limited to girls only. 

### 4.3. Strengths and Weaknesses

We have been able to test for an association between the green space availability in early life and cognitive aging between childhood and later life, after the adjustment for childhood and adulthood covariates, due to the rich longitudinal information collected on the participants from the LBC1936. Our use of a publicly accessible measure of green space was a strength, in that public parks would have provided opportunities for physical activity and stress reduction during childhood, arguably more so than ambient greenness that includes street trees, for example [37]; however, we acknowledge that other green spaces (e.g., golf courses) may have been used and may have benefited the participants. Our work is unique, in that we are attempting to recreate the activity spaces of the children in the 1940s. The landscape available to children has changed appreciably since then; no doubt a much higher percentage of children were undertaking unaccompanied travel in the 1940s compared with today. Estimates of such independent travel to school from 1971 and 1990 show that, in 1971, 80% of seven and eight-year-olds (data unavailable for older age groups) traveled to school without adult supervision, compared with only 9% in 1990 [38]. During this time, the volume of road traffic doubled but fatal accidents involving children halved [38]. Therefore, a child’s activity space in the 1940s would have been less constrained and more influenced by factors such as the RTA density than today. The activity space indices created in our current study are useful and valid for exploring a more complete exposure to green space in early life than those focused solely on the area around the home. 

However, there were limitations to the study. We have focused on childhood and adolescence but the park availability during pregnancy and in infancy may be equally as important; these aspects were not measured in the current study. For the residential information, we were limited by the retrospective data collection of childhood residential information, which is prone to recall bias. Due to the sample criteria (e.g., living in Edinburgh throughout life), the sample may have suffered from selection bias, although we have previously shown that this Edinburgh ‘life-course’ sample does not deviate substantially from the full sample on key characteristics [1]. For the two time periods of interest (childhood and adolescence), we only had two locations to determine the activity space indices. These indices were limited by the simple estimation of the time spent in each location (based on a typical weekday), which doesn’t account for weekend days or holidays when the exposure to the green space surrounding the home would arguably be much more important than the exposure surrounding the school [39]. The optimal routes, in particular, may be prone to error, as it is impossible to know how precisely they represent the actual route taken by the participant, which might have varied day-to-day [40]. We are also limited by the use of the contemporary road networks in estimating the movement in the city during the 1950s. However, there were very limited changes to the urban infrastructure in Edinburgh compared to other cities (e.g., Glasgow’s introduction of inner-city motorways), due to greater local support for conserving the city’s architectural heritage and the smaller scale of any slum clearance projects. An important determinant of the route is the mode of transport, which was assumed to be walking, as this would have been more likely during the 1940s; however, we are unable to discount that this may have drawn associations towards the null, as argued elsewhere [41]. Finally, when analyzing geographic-based exposures, there is debate as to whether a correction for multiple testing should be applied when using multiple buffers.

### 4.4. Study Implications

The access to nearby green space at an adolescent age near home, school, and between the two, may be an important factor for cognitive wellbeing that remains apparent into old age. We found this association in a generation who would have had considerably greater freedom to roam at that age than the current adolescents do. The road characteristics (i.e., the frequency of traffic accidents) also influenced the degree to which this benefit was found. 

These findings are particularly relevant today, as children are spending more time indoors [42] and the activity spaces of children and adolescents, especially girls, are increasingly constrained, both spatially and socially (i.e., adult supervised) [20]. Unstructured play and exploration can deliver the benefits of green space for young people [43]; however, this relies on the design of urban environments that are safe, both structurally and as perceived. A recent systematic review found some evidence for interventions on road traffic safety being associated with reduced injuries, casualties, and collisions involving school children [44]. Policymakers should look to implement similar interventions and work with researchers using data on salutogenic areas (e.g., parks and green walking and cycling infrastructure) to promote safe and healthy spaces and routes through the city.

Future research should address the challenge of incorporating activity space-based exposure over the life-course. In addition to age, sex, and co-existing environmental variables, future studies should investigate socioeconomic modifiers of lifetime activity space extent. The main challenge is to determine ways to gather and process the relevant data. In historical cohorts, additional questions on the places frequented for recreation, e.g., the locations of friends and family [45], could be added to life grid questionnaires (although care would have to be taken to avoid respondent fatigue). In contemporary cohorts, a series of GPS collections could also be taken longitudinally using GPS loggers, as in the Adolescent Health and Development in Context study [46], or perhaps less intrusively via smartphone apps [47]. The information from the apps could be processed using algorithms to determine local activity spaces [48], providing data on where the participants move in relation to the set buffer zones and how the optimal routes compare to the actual routes taken [49]. 

## 5. Conclusions

Utilizing information on everyday locations supplementary to the home to determine the public park availability in early life has reinforced previous associations with cognitive aging. Factors such as road traffic accidents seem to be important in determining the size of an adolescent’s activity space and their propensity to spend time in natural environments, which may ultimately promote or inhibit their successful cognitive aging later in life. Our study has demonstrated the value of integrating activity space measures into life-course analyses, which is an important priority for researchers concerned with the connections between health and place. 

## Figures and Tables

**Figure 1 ijerph-16-00632-f001:**
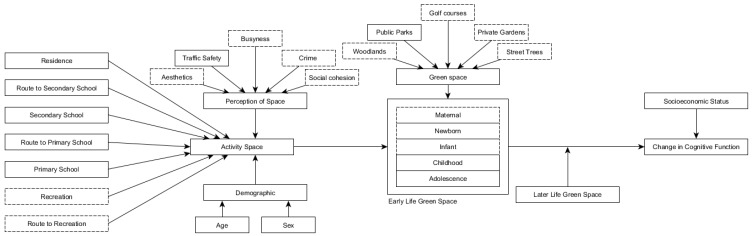
The conceptual model of the relationship between early life green space and cognitive aging in later life. The continuous lined boxes: Measured in the current study. The dotted lined boxes: Unmeasured in the current study.

**Figure 2 ijerph-16-00632-f002:**
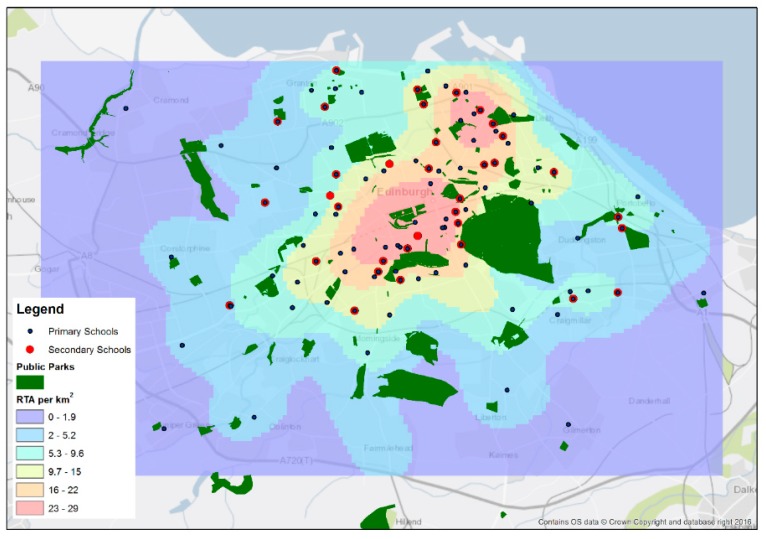
Public parks (1949) and the road traffic accident density (1937/1946) in Edinburgh, UK, in relation to primary and secondary schools.

**Figure 3 ijerph-16-00632-f003:**
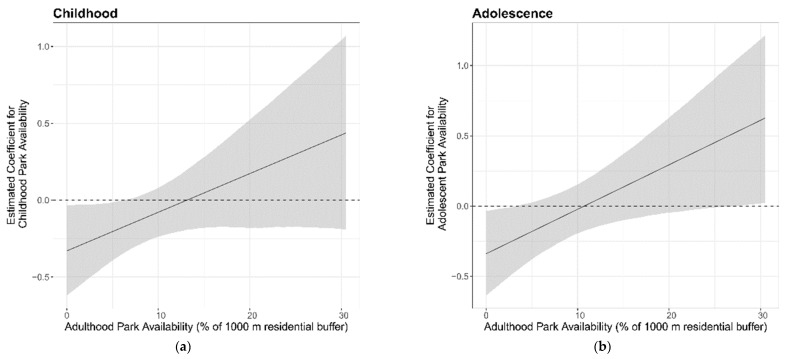
The marginal effects of the activity space park availability on cognitive change from age 70–76, by age period: (**a**) Childhood (age 4–11); (**b**) Adolescence (age 11–18).

**Table 1 ijerph-16-00632-t001:** Selected characteristics for the LBC1936 sample (N = 281).

Characteristic	Mean (±SD); N (%)
Sex	
Female	134 (48)
Father’s Occupational Social Class	
Professional-managerial (I/II)	62 (22)
Skilled, partly skilled, unskilled (III/IV/V)	203 (72)
Missing	16 (6)
Participant’s Occupational Social Class	
Professional-managerial (I/II)	151 (54)
Skilled, partly skilled, unskilled (III/IV/V)	127 (45)
Missing	3 (1)
Public parks (%)	
Childhood Index	8.6 ± 7.3
Adolescence Index	9.1 ± 6.9
Road traffic accident density (per km^2^)	
Childhood Index	6.9 ± 3.6
Adolescence Index	14.5 ± 5.2
Change in cognitive function from age 70 to age 76 on Moray House Test	1.01 ± 0.95
Missing	0

Notes: For continuous variables, the mean is presented with the standard deviation. For categorical variables, the number is presented with the percentage in brackets. The percentages may not add to 100 due to rounding.

**Table 2 ijerph-16-00632-t002:** The life-course analysis on childhood (age 4–11) and adolescent (age 11–18) park availability and cognitive change in later life.

Life-course Park Availability ^a^	Change in Cognitive Function from Age 70 to Age 76 ^b^				
	All ^c^	Males ^d^	Females ^d^	Low Traffic Accident Density ^c^	High Traffic Accident Density ^c^
Childhood Activity Space * Adulthood Residence	0.22 (−0.07 to 0.51) [0.1475]	0.13 (−0.32 to 0.57) [0.5764]	0.33 (−0.07 to 0.73) [0.1016]	0.52 (−0.08 to 1.13) [0.0877]	0.14 (−0.27 to 0.54) [0.5054]
Adolescent Activity Space * Adulthood Residence	0.27 (0.00 to 0.55) [0.0496]	0.21 (−0.20 to 0.62) [0.3100]	0.33 (−0.07 to 0.72) [0.1022]	0.98 (0.36 to 1.60) [0.0022]	0.08 (−0.29 to 0.45) [0.6677]

^a^ The park availability is determined using the % of the area within a 1000 m buffer surrounding the home and school and a 200 m buffer surrounding the route to school. ^b^ Odds Ratio (95% CI) [*p*-value]. ^c^ Adjusted for the sex, father’s occupational social class, people per room in the childhood home, childhood smoking, adulthood OSC, alcohol consumption, and smoking status. ^d^ Adjusted for the father’s occupational social class, people per room in the childhood home, childhood smoking, adulthood OSC, alcohol consumption, and smoking status. * Interaction term.

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
