# Peer review of "Association Between the Activity Space Exposure to Parks in Childhood and Adolescence and Cognitive Aging in Later Life"

_ijerph, 2019, doi:10.3390/ijerph16040632_

Round 1
Reviewer 1 Report
Line 82: Change "hypothesised" to "hypothesized"
Line 106: Change "minimises" to "minimizes"
Line 113, 114: Change "digitised" to"digitized"
Line 120: Change "hypothesised" to "hypothesized"
Please do the same for the words "standardised" and "dichotomised"
Line 131: Add comma after the word "availability"
Line 175: Add comma after the "Therefore"
Author Response
Point 1-7:
Line 82: Change "hypothesised" to "hypothesized"
Line 106: Change "minimises" to "minimizes"
Line 113, 114: Change "digitised" to"digitized"
Line 120: Change "hypothesised" to "hypothesized"
Please do the same for the words "standardised" and "dichotomised"
Line 131: Add comma after the word "availability"
Line 175: Add comma after the "Therefore"
Response 1-7: We have made the changes to spelling and grammar that were suggested by the reviewer.
Reviewer 2 Report
Major alterations
Not applicable.
Minor alterations
Line 37 – Please add “to green spaces” in the sentence “…association between higher exposure to green spaces…”.
Line 55 – I would recommend you to insert “age” in the sentence “… cognitive function from age 70 to 76…”. Without specifying units one can think that those numbers are related with the measure of lifelong exposure to green spaces that you mention previously.
Line 98– Please review the sentence. In my opinion, you should remove “who” from the sentence and only keep “whom”. That is, “…most of whom took a test…”.
Line 167 – Please follow the same recommendations mentioned on line 55, as presenting those numbers without units might be misleading (related with the score of the MHT).
Author Response
Point 1: Line 37 – Please add “to green spaces” in the sentence “…association between higher exposure to green spaces…”.
Response Point 1: We have made the suggested edit to the sentence.
Point 2: Line 55 – I would recommend you to insert “age” in the sentence “… cognitive function from age70 to 76…”. Without specifying units one can think that those numbers are related with the measure of lifelong exposure to green spaces that you mention previously.
Reponse Point 2: We agree that the addition of age makes it clearer for the reader and have made the suggested edit to the sentence.
Point 3: Line 98– Please review the sentence. In my opinion, you should remove “who” from the sentence and only keep “whom”. That is, “…most of whom took a test…”.
Response Point 3: Yes, we agree with the reviewer that 'who' is redundant in this sentence and have removed it.
Point 4: Line 167 – Please follow the same recommendations mentioned on line 55, as presenting those numbers without units might be misleading (related with the score of the MHT).
Response Point 4: As with point 2, we agree that specifying that 70 to 76 relates to age is a useful addition to the sentence. We have also changed any other instance of this in the rest of the manuscript.
Reviewer 3 Report
The authors addressed an important public health issue using a novel approach. They have made a great effort incorporating most of the concerns that the reviewers pointed out in the previous submission. Especially, they have clarified important parts of the conceptual model. Thus, I will recommend the publication of the paper after minor changes.
Abstract
I will suggest the authors to include more results in the abstract, including those that are not possitive.
Results
Please, try to avoid the term "significant" in the explanation of the results.
Author Response
Point 1: The authors addressed an important public health issue using a novel approach. They have made a great effort incorporating most of the concerns that the reviewers pointed out in the previous submission. Especially, they have clarified important parts of the conceptual model. Thus, I will recommend the publication of the paper after minor changes.
Response Point 1: We thank the reviewer for their positive feedback on the value of the research and the effort by the authors to incorporate the concerns of the reviewers in the resubmitted manuscript.
Point 2: I will suggest the authors to include more results in the abstract, including those that are not possitive.
Response Point 2: We agree with the reviewer that it is useful to include futher results in the abstract. We have added the non-positive compliment to the main result on ln 28, showing that when concurrent RTA density is higher there is no association between green space and cognitive ageing:
Point 3: Please, try to avoid the term "significant" in the explanation of the results.
Response Point 3: We agree that the term 'significant' is not always helpful in the description of the results. We have removed the words 'non-significant' on ln 47 and changed the sentence to :
Availability of parks during adolescence was associated with better cognitive ageing (β=0.27, 95% CI 0.00 to 0.55), to a greater extent compared with childhood availability of parks (β=0.22, 95% CI -0.07 to 0.51) (Table 2).
We have changed 'non-significant associations' on ln 55 to:
"The park availability during adolescence had strong associations with better cognitive ageing in participants with lower RTA density (β=0.98, 95% CI 0.36 to 1.60); this was not the casewhen RTA density was higher (β=0.08, 95% CI -0.29 to 0.45) (Table 2)."
We changed 'no significant associations' on ln 57 to:
We change
In the sensitivity analysis with smaller and larger sized buffers, childhood or adolescent park availability were not associated with cognitive ageing (Table S2).